# High-dimensional neural spike train analysis with generalized count linear dynamical systems

**Yuanjun Gao**
Department of Statistics
Columbia University
New York, NY 10027
yg2312@columbia.edu

**Lars Buesing**
Department of Statistics
Columbia University
New York, NY 10027
lars@stat.columbia.edu

**Krishna V. Shenoy**
Department of Electrical Engineering
Stanford University
Stanford, CA 94305
shenoy@stanford.edu

**John P. Cunningham**
Department of Statistics
Columbia University
New York, NY 10027
jpc2181@columbia.edu

## Abstract

Latent factor models have been widely used to analyze simultaneous recordings of spike trains from large, heterogeneous neural populations. These models assume the signal of interest in the population is a low-dimensional latent intensity that evolves over time, which is observed in high dimension via noisy point-process observations. These techniques have been well used to capture neural correlations across a population and to provide a smooth, denoised, and concise representation of high-dimensional spiking data. One limitation of many current models is that the observation model is assumed to be Poisson, which lacks the flexibility to capture under- and over-dispersion that is common in recorded neural data, thereby introducing bias into estimates of covariance. Here we develop the generalized count linear dynamical system, which relaxes the Poisson assumption by using a more general exponential family for count data. In addition to containing Poisson, Bernoulli, negative binomial, and other common count distributions as special cases, we show that this model can be tractably learned by extending recent advances in variational inference techniques. We apply our model to data from primate motor cortex and demonstrate performance improvements over state-of-the-art methods, both in capturing the variance structure of the data and in held-out prediction.

## 1 Introduction

Many studies and theories in neuroscience posit that high-dimensional populations of neural spike trains are a noisy observation of some underlying, low-dimensional, and time-varying signal of interest. As such, over the last decade researchers have developed and used a number of methods for jointly analyzing populations of simultaneously recorded spike trains, and these techniques have become a critical part of the neural data analysis toolkit [1]. In the supervised setting, generalized linear models (GLM) have used stimuli and spiking history as covariates driving the spiking of the neural population [2, 3, 4, 5]. In the unsupervised setting, latent variable models have been used to extract low-dimensional hidden structure that captures the variability of the recorded data, both temporally and across the population of neurons [6, 7, 8, 9, 10, 11].

In both these settings, however, a limitation is that spike trains are typically assumed to be conditionally Poisson, given the shared signal [8, 10, 11]. The Poisson assumption, while offering algorithmic conveniences in many cases, implies the property of equal dispersion: the conditional mean and variance are equal. This well-known property is particularly troublesome in the analysis of neural spike trains, which are commonly observed to be either over- or under-dispersed [12] (variance greater than or less than the mean). No doubly stochastic process with a Poisson observation can capture under-dispersion, and while such a model can capture over-dispersion, it must do so at the cost of erroneously attributing variance to the latent signal, rather than the observation process.

To allow for deviation from the Poisson assumption, some previous work has instead modeled the data as Gaussian [7] or using more general renewal process models [13, 14, 15]; the former of which does not match the count nature of the data and has been found inferior [8], and the latter of which requires costly inference that has not been extended to the population setting. More general distributions like the negative binomial have been proposed [16, 17, 18], but again these families do not generalize to cases of under-dispersion. Furthermore, these more general distributions have not yet been applied to the important setting of latent variable models.

Here we employ a count-valued exponential family distribution that addresses these needs and includes much previous work as special cases. We call this distribution the *generalized count* (GC) distribution [19], and we offer here four main contributions: *(i)* we introduce the GC distribution and derive a variety of commonly used distributions that are special cases, using the GLM as a motivating example (§2); *(ii)* we combine this observation likelihood with a latent linear dynamical systems prior to form a GC linear dynamical system (GCLDS; §3); *(iii)* we develop a variational learning algorithm by extending the current state-of-the-art methods [20] to the GCLDS setting (§3.1); and *(iv)* we show in data from the primate motor cortex that the GCLDS model provides superior predictive performance and in particular captures data covariance better than Poisson models (§4).

## 2 Generalized count distributions

We define the generalized count distribution as the family of count-valued probability distributions:

$$p_{\mathcal{GC}}(k; \theta, g(\cdot)) = \frac{\exp(\theta k + g(k))}{k! M(\theta, g(\cdot))}, \quad k \in \mathbb{N} \tag{1}$$

where $\theta \in \mathbb{R}$ and the function $g : \mathbb{N} \to \mathbb{R}$ parameterizes the distribution, and $M(\theta, g(\cdot)) = \sum_{k=0}^{\infty} \frac{\exp(\theta k + g(k))}{k!}$ is the normalizing constant. The primary virtue of the GC family is that it recovers all common count-valued distributions as special cases and naturally parameterizes many common supervised and unsupervised models (as will be shown); for example, the function $g(k) = 0$ implies a Poisson distribution with rate parameter $\lambda = \exp\{\theta\}$. Generalizations of the Poisson distribution have been of interest since at least [21], and the paper [19] introduced the GC family and proved two additional properties: first, that the expectation of any GC distribution is monotonically increasing in $\theta$, for a fixed $g(k)$; and second – and perhaps most relevant to this study – concave (convex) functions $g(\cdot)$ imply under-dispersed (over-dispersed) GC distributions. Furthermore, often desired features like zero truncation or zero inflation can also be naturally incorporated by modifying the $g(0)$ value [22, 23]. Thus, with $\theta$ controlling the (log) rate of the distribution and $g(\cdot)$ controlling the "shape" of the distribution, the GC family provides a rich model class for capturing the spiking statistics of neural data. Other discrete distribution families do exist, such as the Conway-Maxwell-Poisson distribution [24] and ordered logistic/probit regression [25], but the GC family offers a rich exponential family, which makes computation somewhat easier and allows the $g(\cdot)$ functions to be interpreted.

Figure 1 demonstrates the relevance of modeling dispersion in neural data analysis. The left panel shows a scatterplot where each point is an individual neuron in a recorded population of neurons from primate motor cortex (experimental details will be described in §4). Plotted are the mean and variance of spiking activity of each neuron; activity is considered in 20ms bins. For reference, the equi-dispersion line implied by a homogeneous Poisson process is plotted in red, and note further that all doubly stochastic Poisson models would have an implied dispersion *above* this Poisson line. These data clearly demonstrate meaningful under-dispersion, underscoring the need for the present advance. The right panel demonstrates the appropriateness of the GC model class, showing that a convex/linear/concave function $g(k)$ will produce the expected over/equal/under-dispersion. Given

the left panel, we expect under-dispersed GC distributions to be most relevant, but indeed many neural datasets also demonstrate over and equi-dispersion [12], highlighting the need for a flexible observation family.

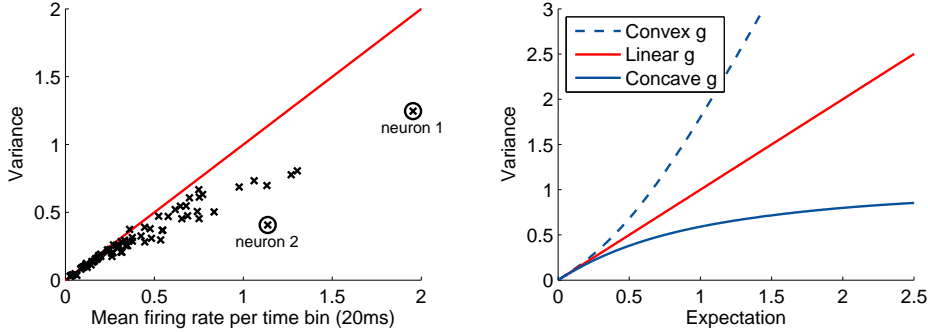

Figure 1: *Left panel*: mean firing rate and variance of neurons in primate motor cortex during the peri-movement period of a reaching experiment (see §4). The data exhibit under-dispersion, especially for high firing-rate neurons. The two marked neurons will be analyzed in detail in Figure 2. *Right panel*: the expectation and variance of the GC distribution with different choices of the function $g$

To illustrate the generality of the GC family and to lay the foundation for our unsupervised learning approach, we consider briefly the case of supervised learning of neural spike train data, where generalized linear models (GLM) have been used extensively [4, 26, 17]. We define GCGLM as that which models a single neuron with count data $y_i \in \mathbb{N}$, and associated covariates $x_i \in \mathbb{R}^p (i = 1, ..., n)$ as

$$y_i \sim \mathcal{GC}(\theta(x_i), g(\cdot)), \quad \text{where} \quad \theta(x_i) = x_i\beta. \tag{2}$$

Here $\mathcal{GC}(\theta, g(\cdot))$ denotes a random variable distributed according to (1), $\beta \in \mathbb{R}^p$ are the regression coefficients. This GCGLM model is highly general. Table 1 shows that many of the commonly used count-data models are special cases of GCGLM, by restricting the $g(\cdot)$ function to have certain parametric form. In addition to this convenient generality, one benefit of our parametrization of the GC model is that the curvature of $g(\cdot)$ directly measures the extent to which the data deviate from the Poisson assumption, allowing us to meaningfully interrogate the form of $g(\cdot)$. Note that (2) has no intercept term because it can be absorbed in the $g(\cdot)$ function as a linear term $\alpha k$ (see Table 1).

Unlike previous GC work [19], our parameterization implies that maximum likelihood parameter estimation (MLE) is a tractable convex program, which can be seen by considering:

$$(\hat{\beta}, \hat{g}(\cdot)) = \arg \max_{(\beta, g(\cdot))} \sum_{i=1}^{n} \log p(y_i) = \arg \max_{(\beta, g(\cdot))} \sum_{i=1}^{n} \left[ (x_i\beta)y_i + g(y_i) - \log M(x_i\beta, g(\cdot)) \right]. \tag{3}$$

First note that, although we have to optimize over a function $g(\cdot)$ that is defined on all non-negative integers, we can exploit the empirical support of the distribution to produce a finite optimization problem. Namely, for any $k^*$ that is not achieved by any data point $y_i$ (i.e., the count $\#\{i|y_i = k^*\} = 0$), the MLE for $g(k^*)$ must be $-\infty$, and thus we only need to optimize $g(k)$ for $k$ that have empirical support in the data. Thus $g(k)$ is a finite dimensional vector. To avoid the potential overfitting caused by truncation of $g_i(\cdot)$ beyond the empirical support of the data, we can enforce a large (finite) support and impose a quadratic penalty on the second difference of g(.), to encourage linearity in $g(\cdot)$ (which corresponds to a Poisson distribution). Second, note that we can fix $g(0) = 0$ without loss of generality, which ensures model identifiability. With these constraints, the remaining $g(k)$ values can be fit as free parameters or as convex-constrained (a set of linear inequalities on $g(k)$; similarly for concave case). Finally, problem convexity is ensured as all terms are either linear or linear within the log-sum-exp function $M(\cdot)$, leading to fast optimization algorithms [27].

## 3 Generalized count linear dynamical system model

With the GC distribution in hand, we now turn to the unsupervised setting, namely coupling the GC observation model with a latent, low-dimensional dynamical system. Our model is a generalization

Table 1: Special cases of GCGLM. For all models, the GCGLM parametrization for $\theta$ is only associated with the slope $\theta(x) = \beta x$, and the intercept $\alpha$ is absorbed into the $g(\cdot)$ function. In all cases we have $g(k) = -\infty$ outside the stated support of the distribution. Whenever unspecified, the support of the distribution and the domain of the $g(\cdot)$ function are non-negative integers $\mathbb{N}$.

| Model Name | Typical Parameterization | GCGLM Parametrization |
|---|---|---|
| Logistic regression (e.g. [25]) | $P(y=k) = \dfrac{\exp\left(k(\alpha + x\beta)\right)}{1 + \exp(\alpha + x\beta)}$ | $g(k) = \alpha k; k = 0, 1$ |
| Poisson regression (e.g., [4, 26] ) | $P(y=k) = \dfrac{\lambda^k}{k!}\exp(-\lambda);$ $\lambda = \exp(\alpha + x\beta)$ | $g(k) = \alpha k$ |
| Adjacent category regression (e.g., [25] ) | $\dfrac{P(y=k+1)}{P(y=k)} = \exp(\alpha_k + x\beta)$ | $g(k) = \sum_{i=1}^{k}(\alpha_{i-1} + \log i);$ $k = 0, 1, ..., K$ |
| Negative binomial regression (e.g., [17, 18]) | $P(y=k) = \dfrac{(k+r-1)!}{k!(r-1)!}(1-p)^r p^k$ $p = \exp(\alpha + x\beta)$ | $g(k) = \alpha k + \log(k+r-1)!$ |
| COM-Poisson regression (e.g., [24]) | $P(y=k) = \dfrac{\lambda^k}{(k!)^\nu}\Big/\sum_{j=1}^{+\infty}\dfrac{\lambda^j}{(j!)^\nu}$ $\lambda = \exp(\alpha + x\beta)$ | $g(k) = \alpha k + (1-\nu)\log k!$ |

of linear dynamical systems with Poisson likelihoods (PLDS), which have been extensively used for analysis of populations of neural spike trains [8, 11, 28, 29]. Denoting $y_{rti}$ as the observed spike-count of neuron $i \in \{1, ..., N\}$ at time $t \in \{1, ..., T\}$ on experimental trial $r \in \{1, ..., R\}$, the PLDS assumes that the spike activity of neurons is a noisy Poisson observation of an underlying low-dimensional latent state $\mathbf{x}_{rt} \in \mathbb{R}^p$, (where $p \ll N$), such that:

$$y_{rti}|\mathbf{x}_{rt} \sim \text{Poisson}\left(\exp\left\{c_i^\top \mathbf{x}_{rt} + \mathbf{d}_i\right\}\right). \tag{4}$$

Here $C = \begin{bmatrix} c_1 & ... & c_N \end{bmatrix}^\top \in \mathbb{R}^{N \times p}$ is the factor loading matrix mapping the latent state $\mathbf{x}_{rt}$ to a log rate, with time and trial invariant baseline log rate $\mathbf{d} \in \mathbb{R}^N$. Thus the vector $C\mathbf{x}_{rt} + \mathbf{d}$ denotes the vector of log rates for trial $r$ and time $t$. Critically, the latent state $\mathbf{x}_{rt}$ can be interpreted as the underlying signal of interest that acts as the "common input signal" to all neurons, which is modeled *a priori* as a linear Gaussian dynamical system (to capture temporal correlations):

$$\begin{aligned} \mathbf{x}_{r1} &\sim \mathcal{N}(\mu_1, Q_1) \\ \mathbf{x}_{r(t+1)}|\mathbf{x}_{rt} &\sim \mathcal{N}(A\mathbf{x}_{rt} + \mathbf{b}_t, Q), \end{aligned} \tag{5}$$

where $\mu_1 \in \mathbb{R}^p$ and $Q_1 \in \mathbb{R}^{p \times p}$ parameterize the initial state. The transition matrix $A \in \mathbb{R}^{p \times p}$ and innovations covariance $Q \in \mathbb{R}^{p \times p}$ parameterize the dynamical state update. The optional term $\mathbf{b}_t \in \mathbb{R}^p$ allows the model to capture a time-varying firing rate that is fixed across experimental trials. The PLDS has been widely used and has been shown to outperform other models in terms of predictive performance, including in particular the simpler Gaussian linear dynamical system [8].

The PLDS model is naturally extended to what we term the generalized count linear dynamical system (GCLDS) by modifying equation (4) using a GC likelihood:

$$y_{rti}|\mathbf{x}_{rt} \sim \mathcal{GC}\left(c_i^\top \mathbf{x}_{rt}, g_i(\cdot)\right). \tag{6}$$

Where $g_i(\cdot)$ is the $g(\cdot)$ function in (1) that models the dispersion for neuron $i$. Similar to the GLM, for identifiability, the baseline rate parameter $\mathbf{d}$ is dropped in (6) and we can fix $g(0) = 0$. As with the GCGLM, one can recover preexisting models, such as an LDS with a Bernoulli observation, as special cases of GCLDS (see Table 1).

### 3.1 Inference and learning in GCLDS

As is common in LDS models, we use expectation-maximization to learn parameters $\Theta = \{A, \{\mathbf{b}_t\}_t, Q, Q_1, \mu_1, \{g_i(\cdot)\}_i, C\}$. Because the required expectations do not admit a closed form

as in previous similar work [8, 30], we required an additional approximation step, which we implemented via a variational lower bound. Here we briefly outline this algorithm and our novel contributions, and we refer the reader to the full details in the supplementary materials.

First, each E-step requires calculating $p(\mathbf{x}_r|\mathbf{y}_r, \Theta)$ for each trial $r \in \{1, ..., R\}$ (the conditional distribution of the latent trajectories $\mathbf{x}_r = \{\mathbf{x}_{rt}\}_{1 \leq t \leq T}$, given observations $\mathbf{y}_r = \{y_{rti}\}_{1 \leq t \leq T, 1 \leq i \leq N}$ and parameter $\Theta$). For ease of notation below we drop the trial index $r$. These posterior distributions are intractable, and in the usual way we make a normal approximation $p(\mathbf{x}|\mathbf{y}, \Theta) \approx q(\mathbf{x}) = \mathcal{N}(\mathbf{m}, V)$. We identify the optimal $(\mathbf{m}, V)$ by maximizing a variational Bayesian lower bound (the so-called evidence lower bound or "ELBO") over the variational parameters $\mathbf{m}, V$ as:

$$
\begin{aligned}
\mathcal{L}(\mathbf{m}, V) =& E_{q(\mathbf{x})}\left[\log\left(\frac{p(\mathbf{x}|\Theta)}{q(\mathbf{x})}\right)\right] + E_{q(\mathbf{x})}[\log p(\mathbf{y}|\mathbf{x}, \Theta)] \quad\quad (7)\\
=& \frac{1}{2}\left(\log|V| - \mathrm{tr}[\Sigma^{-1}V] - (\mathbf{m} - \mu)^T\Sigma^{-1}(\mathbf{m} - \mu)\right) + \sum_{t,i} E_{q(\mathbf{x}_t)}[\log p(y_{ti}|\mathbf{x}_t)] + \mathrm{const},
\end{aligned}
$$

which is the usual form to be maximized in a variational Bayesian EM (VBEM) algorithm [11]. Here $\mu \in \mathbb{R}^{pT}$ and $\Sigma \in \mathbb{R}^{pT \times pT}$ are the expectation and variance of $\mathbf{x}$ given by the LDS prior in (5). The first term of (7) is the negative Kullback-Leibler divergence between the variational distribution and prior distribution, encouraging the variational distribution to be close to the prior. The second term involving the GC likelihood encourages the variational distribution to explain the observations well. The integrations in the second term are intractable (this is in contrast to the PLDS case, where all integrals can be calculated analytically [11]). Below we use the ideas of [20] to derive a tractable, further lower bound. Here the term $E_{q(\mathbf{x}_t)}[\log p(y_{ti}|\mathbf{x}_t)]$ can be reduced to:

$$
\begin{aligned}
E_{q(\mathbf{x}_t)}[\log p(y_{ti}|\mathbf{x}_t)] =& E_{q(\eta_{ti})}\left[\log p_{\mathcal{GC}}(y|\eta_{ti}, g_i(\cdot))\right]\\
=& E_{q(\eta_{ti})}\left[y_{ti}\eta_{ti} + g_i(y_{ti}) - \log y_{ti}! - \log\sum_{k=0}^{K}\frac{1}{k!}\exp(k\eta_{ti} + g_i(k))\right], \quad (8)
\end{aligned}
$$

where $\eta_{ti} = c_i^T\mathbf{x}_t$. Denoting $\nu_{tik} = k\eta_{ti} + g_i(k) - \log(k!) = kc_i^T\mathbf{x}_t + g_i(k) - \log k!$, (8) is reduced to $E_{q(\nu)}[\nu_{tiy_{ti}} - \log(\sum_{0 \leq k \leq K}\exp(\nu_{tik}))]$. Since $\nu_{tik}$ is a linear transformation of $\mathbf{x}_t$, under the variational distribution $\nu_{tik}$ is also normally distributed $\nu_{tik} \sim \mathcal{N}(h_{tik}, \rho_{tik})$. We have $h_{tik} = kc_i^T\mathbf{m}_t + g_i(k) - \log k!, \rho_{tik} = k^2c_i^TV_tc_i$, where $(\mathbf{m}_t, V_t)$ are the expectation and covariance matrix of $\mathbf{x}_t$ under variational distribution. Now we can derive a lower bound for the expectation by Jensen's inequality:

$$
E_{q(\nu_{ti})}\left[\nu_{tiy_{ti}} - \log\sum_{k}\exp(\nu_{tik})\right] \geq h_{tiy_{ti}} - \log\sum_{k=1}^{K}\exp(h_{tik} + \rho_{tik}/2) =: f_{ti}(\mathbf{h}_{ti}, \rho_{ti}). \quad (9)
$$

Combining (7) and (9), we get a tractable variational lower bound:

$$
\mathcal{L}(\mathbf{m}, V) \geq \mathcal{L}^*(\mathbf{m}, V) = E_{q(\mathbf{x})}\left[\log\left(\frac{p(\mathbf{x}|\Theta)}{q(\mathbf{x})}\right)\right] + \sum_{t,i} f_{ti}(\mathbf{h}_{ti}, \rho_{ti}). \quad (10)
$$

For computational convenience, we complete the E-step by maximizing the new evidence lower bound $\mathcal{L}^*$ via its dual [20]. Full details are derived in the supplementary materials.

The M-step then requires maximization of $\mathcal{L}^*$ over $\Theta$. Similar to the PLDS case, the set of parameters involving the latent Gaussian dynamics $(A, \{\mathbf{b}_t\}_t, Q, Q_1, \mu_1)$ can be optimized analytically [8]. Then, the parameters involving the GC likelihood $(C, \{g_i\}_i)$ can be optimized efficiently via convex optimization techniques [27] (full details in supplementary material).

In practice we initialize our VBEM algorithm with a Laplace-EM algorithm, and we initialize each E-step in VBEM with a Laplace approximation, which empirically gives substantial runtime advantages, and always produces a sensible optimum. With the above steps, we have a fully specified learning and inference algorithm, which we now use to analyze real neural data. Code can be found at `https://bitbucket.org/mackelab/pop_spike_dyn`.

## 4 Experimental results

We analyze recordings of populations of neurons in the primate motor cortex during a reaching experiment (`G20040123`), details of which have been described previously [7, 8]. In brief, a rhesus macaque monkey executed 56 cued reaches from a central target to 14 peripheral targets. Before the subject was cued to move (the *go* cue), it was given a preparatory period to plan the upcoming reach. Each trial was thus separated into two temporal epochs, each of which has been suggested to have their own meaningful dynamical structure [9, 31]. We separately analyze these two periods: the preparatory period (1200ms period preceding the go cue), and the reaching period (50ms before to 370ms after the movement onset). We analyzed data across all 14 reach targets, and results were highly similar; in the following for simplicity we show results for a single reaching target (one 56 trial dataset). Spike trains were simultaneously recorded from 96 electrodes (using a Blackrock multi-electrode array). We bin neural activity at 20ms. To include only units with robust activity, we remove all units with mean rates less than 1 spike per second on average, resulting in 81 units for the preparatory period, and 85 units for the reaching period. As we have already shown in Figure 1, the reaching period data are strongly under-dispersed, even absent conditioning on the latent dynamics (implying further under-dispersion in the observation noise). Data during the preparatory period are particularly interesting due to its clear cross-correlation structure.

To fully assess the GCLDS model, we analyze four LDS models – *(i)* GCLDS-full: a separate function $g_i(\cdot)$ is fitted for each neuron $i \in \{1, ..., N\}$; *(ii)* GCLDS-simple: a single function $g(\cdot)$ is shared across all neurons (up to a linear term modulating the baseline firing rate); *(iii)* GCLDS-linear: a truncated linear function $g_i(\cdot)$ is fitted, which corresponds to truncated-Poisson observations; and *(iv)* PLDS: the Poisson case is recovered when $g_i(\cdot)$ is a linear function on all nonnegative integers. In all cases we use the learning and inference of §3.1. We initialize the PLDS using nuclear norm minimization [10], and initialize the GCLDS models with the fitted PLDS. For all models we vary the latent dimension $p$ from 2 to 8.

To demonstrate the generality of the GCLDS and verify our algorithmic implementation, we first considered extensive simulated data with different GCLDS parameters (not shown). In all cases GCLDS model outperformed PLDS in terms of negative log-likelihood (NLL) on test data, with high statistical significance. We also compared the algorithms on PLDS data and found very similar performance between GCLDS and PLDS, implying that GCLDS does not significantly overfit, despite the additional free parameters and computation due to the $g(\cdot)$ functions.

**Analysis of the reaching period.** Figure 2 compares the fits of the two neural units highlighted in Figure 1. These two neurons are particularly high-firing (during the reaching period), and thus should be most indicative of the differences between the PLDS and GCLDS models. The left column of Figure 2 shows the fitted $g(\cdot)$ functions the for four LDS models being compared. It is apparent in both the GCLDS-full and GCLDS-simple cases that the fitted $g$ function is concave (though it was not constrained to be so), agreeing with the under-dispersion observed in Figure 1.

The middle column of Figure 2 shows that all four cases produce models that fit the mean activity of these two neurons very well. The black trace shows the empirical mean of the observed data, and all four lines (highly overlapping and thus not entirely visible) follow that empirical mean closely. This result is confirmatory that the GCLDS matches the mean and the current state-of-the-art PLDS.

More importantly, we have noted the key feature of the GCLDS is matching the dispersion of the data, and thus we expect it should outperform the PLDS in fitting variance. The right column of Figure 2 shows this to be the case: the PLDS significantly overestimates the variance of the data. The GCLDS-full model tracks the empirical variance quite closely in both neurons. The GCLDS-linear result shows that only adding truncation does not materially improve the estimate of variance and dispersion: the dotted blue trace is quite far from the true data in black, and indeed it is quite close to the Poisson case. The GCLDS-simple still outperforms the PLDS case, but it does not model the dispersion as effectively as the GPLDS-full case where each neuron has its own dispersion parameter (as Figure 1 suggests). The natural next question is whether this outperformance is simply in these two illustrative neurons, or if it is a population effect. Figure 3 shows that indeed the population is much better modeled by the GCLDS model than by competing alternatives. The left and middle panels of Figure 3 show leave-one-neuron-out prediction error of the LDS models. For each reaching target we use 4-fold cross-validation and the results are averaged across all 14 reaching

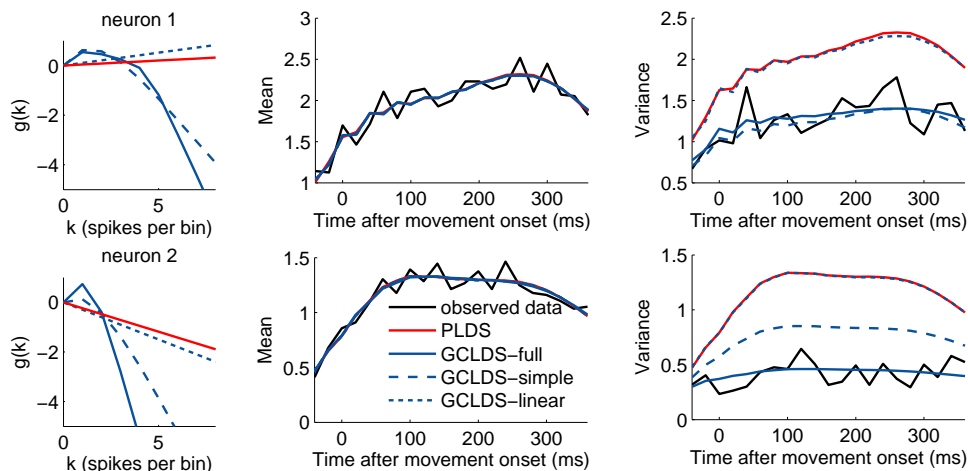

Figure 2: Examples of fitting result for selected high-firing neurons. Each row corresponds to one neuron as marked in left panel of Figure 1 – *left column*: fitted $g(\cdot)$ using GCLDS and PLDS; *middle and right column*: fitted mean and variance of PLDS and GCLDS. See text for details.

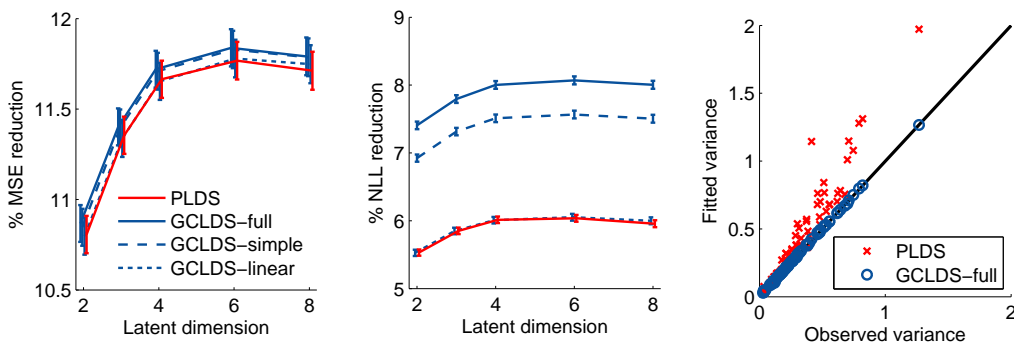

Figure 3: Goodness-of-fit for monkey data during the reaching period – *left panel*: percentage reduction of mean-squared-error (MSE) compared to the baseline (homogeneous Poisson process); *middle panel*: percentage reduction of predictive negative log likelihood (NLL) compared to the baseline; *right panel*: fitted variance of PLDS and GCLDS for all neurons compared to the observed data. Each point gives the observed and fitted variance of a single neuron, averaged across time.

targets. Critically, these predictions are made for all neurons in the population. To give informative performance metrics, we defined baseline performance as a straightforward, homogeneous Poisson process for each neuron, and compare the LDS models with the baseline using percentage reduction of mean-squared-error and negative log likelihood (thus higher error reduction numbers imply better performance). The mean-squared-error (MSE; left panel) shows that the GCLDS offers a minor improvement (reduction in MSE) beyond what is achieved by the PLDS. Though these standard error bars suggest an insignificant result, a paired t-test is indeed significant ($p < 10^{-8}$). Nonetheless this minor result agrees with the middle column of Figure 2, since predictive MSE is essentially a measurement of the mean.

In the middle panel of Figure 3, we see that the GCLDS-full significantly outperforms alternatives in predictive log likelihood across the population ($p < 10^{-10}$, paired t-test). Again this largely agrees with the implication of Figure 2, as negative log likelihood measures both the accuracy of mean and variance. The right panel of Figure 3 shows that the GCLDS fits the variance of the data exceptionally well across the population, unlike the PLDS.

**Analysis of the preparatory period.** To augment the data analysis, we also considered the preparatory period of neural activity. When we repeated the analyses of Figure 3 on this dataset, the same results occurred: the GCLDS model produced concave (or close to concave) $g$ functions

and outperformed the PLDS model both in predictive MSE (minority) and negative log likelihood (significantly). For brevity we do not show this analysis here. Instead, we here compare the temporal cross-covariance, which is also a common analysis of interest in neural data analysis [8, 16, 32] and, as noted, is particularly salient in preparatory activity. Figure 4 shows that GCLDS model fits both the temporal cross-covariance (left panel) and variance (right panel) considerably better than PLDS, which overestimates both quantities.

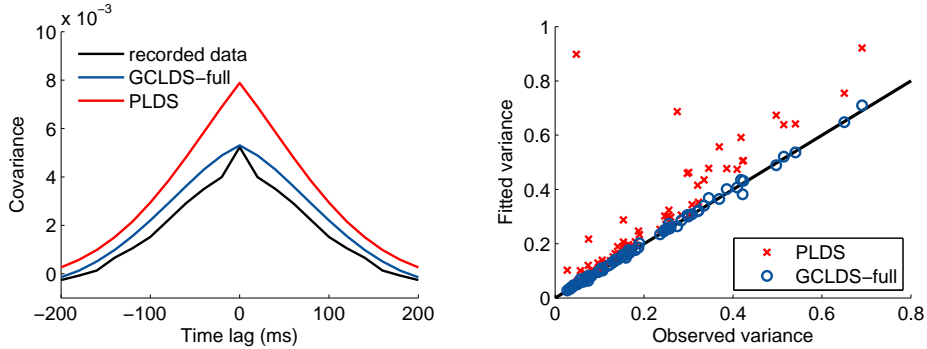

Figure 4: Goodness-of-fit for monkey data during the preparatory period – *Left panel:* Temporal cross-covariance averaged over all $81$ units during the preparatory period, compared to the fitted cross-covariance by PLDS and GCLDS-full. *Right panel*: fitted variance of PLDS and GCLDS-full for all neurons compared to the observed data (averaged across time).

## 5   Discussion

In this paper we showed that the GC family better captures the conditional variability of neural spiking data, and further improves inference of key features of interest in the data. We note that it is straightforward to incorporate external stimuli and spike history in the model as covariates, as has been done previously in the Poisson case [8]. Beyond the GCGLM and GCLDS, the GC family is also extensible to other models that have been used in this setting, such as exponential family PCA [10] and subspace clustering [11]. The cost of this performance, compared to the PLDS, is an extra parameterization (the $g_i(\cdot)$ functions) and the corresponding algorithmic complexity. While we showed that there seems to be no empirical sacrifice to doing so, it is likely that data with few examples and reasonably Poisson dispersion may cause GCLDS to overfit.

### Acknowledgments

JPC received funding from a Sloan Research Fellowship, the Simons Foundation (SCGB#325171 and SCGB#325233), the Grossman Center at Columbia University, and the Gatsby Charitable Trust. Thanks to Byron Yu, Gopal Santhanam and Stephen Ryu for providing the cortical data.

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
