[Supplementary Material · GCLDS_supp.pdf]

# Supplementary material: *High-dimensional neural spike train analysis with generalized count linear dynamical systems*

Here we provide details of the variational inference method for the generalized count linear dynamical system model (GCLDS).

# 1  VBEM algorithm details

## 1.1  Variational Inference in E-step

We first introduce the "vectorized" notation for the GCLDS model. Note that in the E-step the inference is separable across trials, so for ease of notation, we only consider one single trial and drop the trial index $r$. We assume $N$ neurons observed during $T$ time bins. Denote $\mathbf{x}_t$ as the $p$-dimensional latent variable and and $\mathbf{y}_t$ as the $N$-dimensional observation, respectively.

$$\mathbf{x} := \begin{pmatrix} \mathbf{x}_1 \\ \vdots \\ \mathbf{x}_T \end{pmatrix}, \mathbf{y} := \begin{pmatrix} \mathbf{y}_1 \\ \vdots \\ \mathbf{y}_T \end{pmatrix}$$

The prior can be summarized as a multi-variate Gaussian distribution:

$$p(\mathbf{x}) = \mathcal{N}(\mu, \Sigma)$$

where

$$\mu = \begin{pmatrix} \mu_1 \\ A\mu_1 + b_1 \\ \vdots \\ A^{T-1}\mu_1 + \sum_{t=1}^{T-1} A^{T-1-t}b_t \end{pmatrix}, \Sigma^{-1} = \begin{pmatrix} Q_0^{-1} + A^T Q^{-1} A & A^T Q^{-1} \\ Q^{-1}A & Q^{-1} + A^T Q^{-1} A & A^T Q^{-1} \\ & \ddots & \ddots & \ddots \end{pmatrix}.$$

The likelihood has the form

$$p(\mathbf{y}|\mathbf{x}) = \prod_{t,i} p(y_{ti}|\eta_{ti})$$

$$p(y_{ti}|\eta_{ti}) = \mathcal{GC}(y_{ti}|\eta_{ti}, g_i(\cdot))$$

$$\eta := W\mathbf{x}$$

$$W = \text{blk-diag}(C, ..., C),$$

where we stack all the $\eta_{ti}$ in $\eta = (\eta_{11}, ..., \eta_{1N}, ...., \eta_{T1}, ..., \eta_{TN}) \in \mathbb{R}^{NT}$. The log likelihood reads:

$$\log p(\mathbf{x}, \mathbf{y}) \propto -\frac{1}{2}(\mathbf{x} - \mu)^T \Sigma^{-1}(\mathbf{x} - \mu) + \sum_{t,i}[y_{ti}\eta_n + g_i(y_{ti}) - \log(\sum_k \frac{1}{k!}\exp(k\eta_{ti} + g_i(k)))]$$

$$-\sum_{t,i}\log(y_{ti}!) - \frac{1}{2}\log|\Sigma|$$

In the E-step we make a Gaussian approximation to the posterior:

$$p(\mathbf{x}|\mathbf{y}) \approx q(\mathbf{x}) = \mathcal{N}(\mathbf{x}|\mathbf{m}, V).$$

The variational lower bound reads:

$$\mathcal{L}(\mathbf{m}, V) = \int q(\mathbf{x})\log\frac{p(\mathbf{x}, \mathbf{y})}{q(\mathbf{x})}d\mathbf{x}$$

$$= \frac{1}{2}(\log|V| - \text{tr}[\Sigma^{-1}V] - (\mathbf{m} - \mu)^T\Sigma^{-1}(\mathbf{m} - \mu))$$

$$+ \sum_{t,i}E_{q(\eta_{ti})}[\log p(y_{ti}|\eta_{ti})] - \frac{1}{2}\log|\Sigma| + \frac{dT}{2}.$$

Defining $\nu_{tik} = k\eta_{ti} + g_i(k) - \log k!$, we know that $\nu_{tik}$ is also normally distributed under the variational distribution

$$\nu_{tik} \sim \mathcal{N}(h_{tik}, \rho_{tik}).$$

Therefore we can re-write the term $E_{q(x)}[\log p(y_{ti}|\eta_{ti})]$ and find a lower bound of the term by

$$E_{q(\eta_{ti})}\left[\log p(y_{ti}|\eta_{ti})\right]$$

$$= E_{q(\eta_{ti})}\left[y_{ti}\eta_{ti} + g_i(y_{ti}) - \log(y_{ti}!) - \log(\sum_k \frac{1}{k!}\exp(k\eta_{ti} + g_i(k)))\right]$$

$$= E_{q(\nu_{ti})}\left[\nu_{tiy_{ti}} - \log(\sum_{k=0}^{K}\exp(\nu_{tik}))\right]$$

$$\geq h_{tiy_{ti}} - \log(\sum_{k=0}^{K}E_{q(\nu_{ti})}(\exp(\nu_{tik})))$$

$$= h_{tiy_{ti}} - \log(\sum_{k=0}^{K}\exp(h_{nk} + \rho_{nk}/2))$$

where $\nu_{ti} = (\nu_{ti1}, ..., \nu_{tiK})$. We always have $\nu_{ti0} = \rho_{ti0} = 0$. For the other variables define

$$\nu = (\nu_{11}, \nu_{12}, ..., \nu_{1N}, ..., \nu_{T1}, ..., \nu_{TN})^T,$$

and define $\mathbf{h}$ and $\rho$ similarly. We then have the constraints

$$\mathbf{h} := \tilde{W}\mathbf{m} + \tilde{\mathbf{d}}$$

$$\rho := \text{diag}(\tilde{W}V\tilde{W}^T)$$

where

$$\tilde{W} = W \otimes (1, 2, ..., K)^T$$

$$\tilde{\mathbf{d}} = \mathbf{1}_{T \times 1} \otimes (g_1(1) - \log 1!, ..., g_1(K) - \log K!, ...., g_N(1) - \log 1!, ..., g_N(K) - \log K!)^T$$

where $\otimes$ is the Kronecker product. Applying this lower bound and setting $\nu_{ti0} = \rho_{ti0} = 0$, we get the evidence lower bound (ELBO)

$$\mathcal{L}^*(\mathbf{m}, V, \mathbf{h}, \rho) = \frac{1}{2}(\log|V| - \mathrm{tr}[\Sigma^{-1}V] - (\mathbf{m} - \mu)^T \Sigma^{-1}(\mathbf{m} - \mu))$$

$$+ \sum_{t,i}\left[ \mathbf{1}_{\{y_{ti} > 0\}} h_{tiy_{ti}} - \log(1 + \sum_{k=1}^{K} \exp(h_{tik} + \rho_{tik}/2)) \right]$$

the variational inference can now be cast as the optimization problem:

$$\max_{\mathbf{m}, V, \mathbf{h}, \rho} \quad \mathcal{L}^*(\mathbf{m}, V, \mathbf{h}, \rho)$$

$$\text{subject to} \quad V \succeq 0$$

$$\mathbf{h} = \tilde{W}\mathbf{m} + \tilde{\mathbf{d}}$$

$$\rho = \mathrm{diag}(\tilde{W}V\tilde{W}^T)$$

Following [1], we can solve the dual problem

$$\min_{\alpha, \lambda} \max_{\mathbf{m}, V, \mathbf{h}, \rho} L(\mathbf{m}, V, \mathbf{h}, \rho) + \alpha^T(\mathbf{h} - \tilde{W}\mathbf{m} - \tilde{\mathbf{d}}) + \frac{1}{2}\lambda^T(\rho - \mathrm{diag}(\tilde{W}V\tilde{W}^T)),$$

where $\alpha, \lambda \in \mathbb{R}^{TNK}$ are the Lagrange multipliers. The unique maximizer with respect to $(\mathbf{m}, V)$ is given by

$$\mathbf{m}^* = \mu - \Sigma\tilde{W}^T\alpha$$

$$V^* = B_\lambda^{-1} := (\Sigma^{-1} + \tilde{W}^T(\mathrm{diag}\lambda)\tilde{W})^{-1}$$

Maximization over $(\mathbf{h}, \rho)$ is also available in close form. Collecting the term containing $(\mathbf{h}, \rho)$. for $f^*$ to be finite, we need to enforce the constraint $\alpha_{tik} = \lambda_{tik} - \mathbf{1}_{\{y_{ti} = k\}}$. Therefore, we can express everything in terms of $\lambda$

$$f_{ti}^*(\lambda_{ti}) = \max_{\mathbf{h}, \rho} \alpha_{ti}^T \mathbf{h}_{ti} + \lambda_{ti}^T \rho_{ti}/2 + \left[ \mathbf{1}_{\{y_{ti} > 0\}} h_{tiy_{ti}} - \log(1 + \sum_{k=1}^{K} \exp(h_{tik} + \rho_{tik}/2)) \right]$$

$$= \sum_{k=1}^{K} \lambda_{tik} \log \lambda_{tik} + (1 - \sum_{k=1}^{K} \lambda_{tik}) \log(1 - \sum_{k=1}^{K} \lambda_{tik}).$$

Denoting $\tilde{y}_{ti} = (\mathbf{1}_{\{y_{ti}=1\}}, \mathbf{1}_{\{y_{ti}=2\}}, ..., \mathbf{1}_{\{y_{ti}=K\}})$ and $\tilde{y} = (\tilde{y}_{11}, ..., \tilde{y}_{1N}, ..., \tilde{y}_{T1}, ..., \tilde{y}_{TN})$, the dual

problem is reduced to

$$\min_{\lambda} \quad D(\lambda)$$

$$\text{subject to} \quad \lambda_{tik} > 0$$

$$\sum_{k=1}^{K} \lambda_{tik} < 1, \ t = 1, ..., T, n = 1, ..., N, k = 1, ..., K$$

where

$$D(\lambda) := \frac{1}{2}(\lambda - \tilde{y})^T \tilde{W} \Sigma \tilde{W}^T (\lambda - \tilde{y}) - (\tilde{W}\mu + \tilde{\mathbf{d}})^T (\lambda - \tilde{y}) - \frac{1}{2}\log|B_\lambda| + \sum_{t,i} f_{ti}^*(\lambda_{ti})$$

and the gradient of the dual reads

$$D'(\lambda) = \tilde{W}\Sigma\tilde{W}^T(\lambda - \tilde{y}) - \tilde{W}\mu - \tilde{\mathbf{d}} - \frac{1}{2}\text{diag}(WB_\lambda^{-1}W^T) - \sum_n f_{ti}^{*\prime}(\lambda_{ti})$$

## 1.2   M-step

We have two sets of parameters to optimize in the M-step. One set is for the observation $(C, \{g_i(\cdot)\}_i)$, the other is for the dynamical system $(A, \{\mathbf{b}_t\}_t, Q, Q_1, \mu_1)$. It turns out that the M-step can be performed separately for these two sets.

The part of the likelihood about the observation can be written as

$$\mathcal{L}_1(C, g) = \sum_{i=1}^{N} \left[ \sum_{\substack{t=1,...,T \\ r=1,...,R}} y_{rti}(c_i^T m_{rt}) + g_i(y_{rti}) \right.$$

$$\left. - \log(1 + \sum_{k=1}^{K} \frac{1}{k!} \exp(k(c_i^T m_{rt}) + g_i(k) + \frac{1}{2}k^2 c_i^T V_{rt} c_i)) \right]$$

This part is concave and can be optimized efficiently using convex optimization techniques.

The part of the likelihood about the dynamical system has the form

$$\mathcal{L}_2(A, Q, Q_1, \mu_1) = \sum_{r=1}^{R} E_{q(\mathbf{x}_r)} \left[ -\frac{1}{2}(\mathbf{x}_{r1} - \mu_1)^T Q_1^{-1}(\mathbf{x}_{r1} - \mu_1) \right.$$

$$- \frac{1}{2} \sum_{t=1}^{T-1} (\mathbf{x}_{r(t+1)} - A\mathbf{x}_{rt} - \mathbf{b}_t)^T Q^{-1}(\mathbf{x}_{r(t+1)} - A\mathbf{x}_{rt} - \mathbf{b}_t)$$

$$\left. - \frac{1}{2}\log|Q_1| - \frac{T-1}{2}\log|Q| \right]$$

Since everything is quadratic with respect to $\mathbf{x}$, the expectation can be calculated analytically. Moreover, all the parameters can be optimized analytically in close form.

# References

[1] M. Emtiyaz Khan, A. Aravkin, M. Friedlander, and M. Seeger, "Fast dual variational inference for non-conjugate latent gaussian models," in *Proceedings of The 30th International Conference on Machine Learning*, pp. 951–959, 2013.