[Reviews · NeurIPS 2015]

Submitted by Assigned_Reviewer_1

High-dimensional neural spike train analysis with generalized count linear dynamical systems

This paper describes a general exponential-family model (called the "generalized count" (GC) distribution) for multi-neuron spike count data. The model accounts for both under-dispersed and over-dispersed spike count data, and has Poisson, Negative Binomial, Bernoulli, and several other classic models as special cases.

The authors give a clear account of the relationship to other models, and demonstrate the need for a model to capture under-dispersed counts in primate motor cortex.

They then describe an efficient method for maximum-likelihood fitting (and demonstrate concavity of the log-likelihood).

It briefly discusses GC regression before turning to latent variable modeling of population data using a latent Gaussian linear dynamical system and GC observations.

They derive an efficient variational Bayesian inference method and apply the model to data from primate motor cortex, showing that it accounts more accurately for variance and cross-covariance of spike count data, compared to a model with Poisson observations.

The paper is clearly written, original, interesting, and thorough, and likely to have a major impact on the field of neural coding and latent variable modeling of neural data. While the quantitative improvement over existing models is somewhat modest, the model has very nice theoretical properties and is attractive for its ability to account for under- and over-dispersion within a single framework. An outstanding paper -- I expect it to attract considerable interest and attention.

If I have one general criticism it's that it would be nicer to have a clearer idea for how to model single-neuron data, since the "full" model assigns zero probability to all unobserved counts (sending test log-likelihood negative infinity if the test data ever contains a count not observed in training), and the "simple" model requires a full population.

The authors mention this shortcoming briefly in the discussion, but it might be nice to have a simple recommendation for how to handle this (e.g., add a small pseudocount for some reasonable set of possible spike counts).

Technical comments:

106: "this choice has been previously considered and found suboptimal [8]".

No no no, that is not correct! I'm sorry, the Macke et al 2011 paper did NOT examine a fine timescale GLM at all, much less whether it could account for underdispersion (or cross-correlations) in population data. The smallest bin size used was 10ms (which is obviously too big to be effective for modeling short time-scale refractory effects).

So please stop repeating this misleading claim from that earlier paper.

Besides, there's no need for this attack here: this paper addresses an interesting and important problem its own right, whether or not there's some other approach for capturing underdispersion that might also suffice.

(Someone should make a sincere attempt, but it's a topic for another paper.)

Table 1:

If you have space, it might be nice to include the mapping to \theta in the "GCGLM Parametrization", just to make it easier for people to mentally connect the GC model to the classic models.

(e.g., \theta = \log \lambda for Poisson model). Not strictly necessary though if it takes too much space.

eq 6: I found this a little puzzling when I first read it -- might be helpful to spell out what g_i is.

294: "does not significantly overfit"?

This is for how much data -- the same amount as the recording data?

(Presumably there should be substantial overfitting for a small dataset.)

Fig 2: is this all from training data (i.e., just confirming that the model is capable of exhibiting the desired mean-variance relationship), or is there some prediction on test data underlying this figure?
Summary: The paper is clearly written, original, interesting, and thorough, and likely to have a major impact on the field of neural coding and latent variable modeling of neural data. While the quantitative improvement over existing models is somewhat modest, the model has very nice theoretical properties and is attractive for its ability to account for under- and over-dispersion within a single framework. An outstanding paper -- I expect it to attract considerable interest and attention.

Submitted by Assigned_Reviewer_2

The authors develop a learning and inference method for a latent variable model with observations distributed according to a generalized count distribution. Their work extends the Poisson Linear Dynamical System (PLDS) model, allowing for both over- and under-dispersion of observations. The paper is clearly written and very timely. It addresses an important question in modeling neuronal population data - in particular for neural systems where spike counts are under-dispersed, such as in the recording from premotor cortex the authors analyze.

The paper is fairly straightforward. The two main components are the latent dynamics model (LDS) and the generalized count distribution for the observations. Both are individually well understood, but when put together the expectations required for the EM algorithm do not have a closed form. The authors develop a variational approximation, which is the main contribution of the paper. Moreover, they show that the model indeed outperforms the less flexible PLDS model on neural population data from premotor cortex.

I have no major objections, just two comments:

1. The lack of the full model in Fig. 3 (middle) is unfortunate, since Fig. 2 (left) suggests that the g's are quite different between neurons. The authors' explanation that it was omitted because of unobserved count values in the test set is not satisfactory. This problem can be overcome easily by either setting g(k)=g(K) for all k greater than K (where K is the maximum of k in the training set), extrapolating linearly, assuming a simple parametric for for g(k) (e.g. quadratic), etc... I think it's a quite substantial piece of information how variable the g's are between cells and how much this affects the model fit. The authors should try and address this point. Right now it sounds like a bad excuse that had to be made because the problem arose last minute and couldn't be fixed before the submission deadline.

2. While I see the value of having a flexible model allowing under- and over-dispersion of counts, I think the same effect could be achieved by modeling autocorrelations of the response (i.e. history filters for each neuron). Ultimately, in small bins the data have to be Poisson and any deviations from Poisson in larger bins are the result of autocorrelations. The authors comment on this objection, but I find their arguments very weak and not convincing at all. First, I don't agree with the claim that refractory effects were found suboptimal by ref [8]. Ref [8] paper compares GLMs with cross-couplings between neurons to the PLDS model and finds that PLDS performs better in modeling *correlations* between neurons. It does not analyze refractory effects or the dispersion of the counts. Thus, the inferiority argument is not supported. Second, the sentence "it would incorrectly assign all variable dispersion to these terms, rather than to the shape of the count distribution more generally" (line 107) doesn't make much sense to me. What is the "count distribution more generally"? Any deviation of this distribution from Poisson is the result of autocorrelations, which can be learned and modeled. I think this point should be discussed better. Having a flexible model for the count distribution can be very useful for better reasons than those provided by the authors. For instance, it allows one to make the bins larger, which improves efficiency or makes the model even feasible for large datasets.
Summary: A timely and well-written paper introducing a latent variable model that allows for over- and under-dispersed count observations.

Submitted by Assigned_Reviewer_3

Authors propose using a new family of count distribution to model (binned) spike train responses as well as a variational Bayes latent dynamical model using it.

I believe the following recent paper which also solves the underdispersion problem is worth mentioning in this manuscript:

- Koyama, S. (2015). On the spike train variability characterized by Variance-to-Mean power relationship. Neural Computation, 27(7):1530-1548.

Generally GLM family of models with post-spike filters can capture under-dispersion to a certain degree, which is not emphasized enough in this manuscript, misleadingly presenting itself more novel than it is. Also, the comparisons against PLDS are done without the post-spike filters (I presume). Is the improvement still significant when including the post-spike history filters (with a small time scale bins, for example 1 ms or 0.1 ms, and not the 20 ms bin currently used)? Figures (except time-lagged covariance) can still be in 20 ms bins, just the model in a smaller time scale.

I'm psyched to see this come out, and hopefully have a robust open source implementation.

Minor

- I am not convinced as to why neurons with less than 1 spk/s are discarded from analysis.

- The main weakness of the proposed method is the number of parameters in g() and additional approximation in the cost function. However, the results indicate in practice where plenty of data are available, they don't pose serious issues. Note that the Koyama paper has less free parameters, and do not need binning.
Summary: This is a good paper addressing the under/over-dispersion in count models for neural spike trains. It's missing some important citation and comparisons.

Submitted by Assigned_Reviewer_4

I really like this paper, and the possibilities of having a principled alternative to the Poisson are tantalising. There are two things I can think of that the authors can do to potentially turn this into a seminal piece of work. First, can you comment on whether g(.) might have any physiological interpretation, beyond its convexity (which determines dispersion)? At the moment it seems like a mathematical convenience, rather than something meaningful. There's an elevated status to interpretable signals, such as the treatment of spike history in Pillow '05, or how doubly-stochastic Poisson can be thought of as excitability fluctuations (as in Ecker '14 and Goris '14). Answering this question is probably an involved empirical exercise, but simply that it is a question to be answered should be flagged. Second, can you comment on whether g(.) could be used to generalise the Poisson to a multivariate distribution? At the moment, the only reasonable options for modelling spike count correlations are via shared latent variables in a doubly-stochastic generative model, or move to Ising-like joint point process models (with their loss of tractability). There's some evidence in Goris '14 that we need another way. Having suggested this, I'm guessing the convexifying trick of considering only the finite set of observed /k/ values becomes less useful in higher dimensions, and one would be forced to parameterise g(.) beyond the empirical support of the data (with potential loss of convexity in the inference). Perhaps the authors are thinking about this already. If it's dead simple to solve this, say so! If there are more detailed answers forthcoming, I can't wait to hear about them.

Minor comment: the maths would be more readable with a consistent style for vectors/matrices (eg boldface these, to cf scalars).
Summary: The conditionally-Poisson distribution typically used to model spike counts is insufficient. This submission proposes an elegant and very tractable alternative, and does so in a very readable way.

Author Feedback
Author rebuttal: We thank the reviewers for the constructive feedback. A few responses are warranted:

- R1 and R2 both stated a concern that the MLE in the full GCLDS model assigns zero probability to all unobserved counts. Since the submission, we have tackled this issue by enforcing a large support and imposing a quadratic penalty on the second difference of g(.), to encourage linearity in g(.) (which corresponds to a Poisson distribution). Pleasingly, our empirical analysis has shown that this penalty further improves the NLL on held-out data and solves the stated "zero probability" issue (and is robust to regularization coefficients). This improved analysis will be included in the final version.

- R1 and R3 both indicated that the GLM may be improved by including history filters with finer timescale. We agree and will modify our comments, and a more detailed comparison will be interesting. As mentioned by R1, nonetheless we agree that the GCLDS provides a nice framework for modeling dispersion directly and allowing coarser timescale without loss of accuracy.

- R3 stated a concern about overfitting. We agree that the extra parameters g(.) may be prone to overfitting in the small data setting, though we did not encounter any problems in our testing. Nonetheless it is straightforward to impose constraints like concavity/convexity, or a parametric form to the g(.) function, to reduce the number of parameters, without sacrificing its computational convenience.

- R3 noted the Koyama 2015 paper. That paper was not published at the time of submission, but it is an interesting paper that deals with under-dispersion. We will reference it and compare with it.

- R4 stated a concern about g(.) being unconstrained. While a single GC distribution with an arbitrary g(.) function parametrizes all the count distribution, the GCLDS model is well structured and provides desirable flexibility that can not be obtained by PLDS: (1)the parameter theta enables the model to link latent variables (in the LDS case) and/or covariates (in the GLM case) to observations, creating a multivariate dependence; and (2)the extra parameter g(.) provides the model with the flexibility to model the conditionally independent over/under-dispersion that is common in real data. Also, as we have noted above, it is straight-forward to impose finer structure on the g(.) function.

- R7 suggested a physiological interpretation. We have not yet been able to make any solid claims about this, but indeed this is very interesting and important!

- R7 inquired as to a multivariate generalization. While our current model is certainly multivariate (one of the main points of the work), the count distribution is indeed conditionally univariate, and we assume this is what R7 was describing. We have not considered a multivariate generalization, though we agree that is an important area of future work.

- We agree with and will address all minor and technical points in the final version.